# Discordance of Biomarker Expression Profile between Primary Breast Cancer and Synchronous Axillary Lymph Node Metastasis in Preoperative Core Needle Biopsy

**DOI:** 10.3390/diagnostics14030259

**Published:** 2024-01-25

**Authors:** Stefano Marletta, Alexandra Giorlandino, Enrico Cavallo, Michele Dello Spedale Venti, Giorgia Leone, Maria Grazia Tranchina, Lucia Gullotti, Claudia Lucia Bonanno, Graziana Spoto, Giusi Falzone, Irene Tornabene, Carmelina Trovato, Marco Maria Baron, Giuseppe Di Mauro, Lucia Falsaperna, Giuseppe Angelico, Sarah Pafumi, Antonio Rizzo

**Affiliations:** 1Division of Pathology, Humanitas Istituto Clinico Catanese, 95045 Catania, Italy; stefano.marletta92@gmail.com (S.M.); enrico.cavallo@humanitascatania.it (E.C.); michele.dellospedaleventi@humanitascatania.it (M.D.S.V.); giorgia.leone@humanitascatania.it (G.L.); mariagrazia.tranchina@humanitascatania.it (M.G.T.); lucia.gullotti@humanitascatania.it (L.G.); claudia.bonanno@humanitascatania.it (C.L.B.); graziana.spoto@humanitascatania.it (G.S.); giusi.falzone@humanitascatania.it (G.F.); irene.tornabene@humanitascatania.it (I.T.); carmelina.trovato@humanitascatania.it (C.T.); marco.baron@humanitascatania.it (M.M.B.); giuseppe.dimauro@humanitascatania.it (G.D.M.); lucia.falsaperna@humanitascatania.it (L.F.); 2Department of Diagnostics and Public Health, Section of Pathology, University of Verona, 37134 Verona, Italy; 3Pathology Unit, ARNAS Garibaldi Hospital, 95122 Catania, Italy; alexandragiorlandino@hotmail.it; 4Department of Medical, Surgical Sciences and Advanced Technologies G.F. Ingrassia, Anatomic Pathology, University of Catania, 95125 Catania, Italy; giuangel86@hotmail.it; 5Medical Oncology, Humanitas Istituto Clinico Catanese, 95045 Catania, Italy; sarahpafumi@gmail.com; 6Section of Oncology, Department of Medicine, University of Verona, Verona University Hospital Trust (AUOI), 37124 Verona, Italy

**Keywords:** breast cancer, axillary lymph node metastases, biomarkers, immunohistochemistry

## Abstract

**Background**: Breast cancer (BC) is a heterogeneous disease made up of clones with different metastatic potential. Intratumoral heterogeneity may cause metastases to show divergent biomarker expression, potentially affecting chemotherapy response. **Methods**: We investigated the immunohistochemical (IHC) and FISH profile of estrogen receptors (ER), progesterone (PR) receptors, Ki67, and HER2 in a series of BC-matched primary tumors (PTs) and axillary lymph node (ALN) metastases in pre-operative core needle biopsies (CNBs). Phenotypical findings were correlated to morphological features and their clinical implications. **Results**: Divergent expression between PTs and ALNs was found in 10% of the tumors, often involving multiple biomarkers (12/31, 39%). Most (52%) displayed significant differences in ER and PR staining. HER2 divergences were observed in almost three-quarters of the cases (23/31, 74%), with five (16%) switching from negativity to overexpression/amplification in ALNs. Roughly 90% of disparities reflected significant morphological differences between PTs and ALN metastases. Less than half of the discrepancies (12/31, 39%) modified pre/post-operative treatment options. **Conclusions**: We observed relevant discrepancies in biomarker expression between PTs and metastatic ALNs in a noteworthy proportion (10%) of preoperative BC CNBs, which were often able to influence therapies. Hence, our data suggest routine preoperative assessment of biomarkers in both PTs and ALNs in cases showing significant morphological differences.

## 1. Introduction

Breast cancer (BC) remains a significant burden in terms of malignancy incidence and mortality among women worldwide. Despite rising incidence rates, ongoing discoveries about the molecular basis of the disease and the development of specific targeted therapies have led to improved clinical outcomes in recent decades, with a 10-year overall survival of approximately 80% [1]. However, in the vast spectrum of BC presentation, patients’ prognosis is greatly influenced by clinical–pathological variables such as tumor stage, histological type and grade, and therapeutic predictive marker status, among others. Although early-stage tumors (pT1aN0M0, according to the TNM classification) generally display a favorable outcome, with 10-year cancer-related mortality < 5% [2], survival rates significantly drop in patients diagnosed with nodal and distant metastases. In detail, synchronous axillary lymph node (ALN) metastases are acknowledged as one of the most critical parameters predictive of local recurrence and lower overall survival [3,4]. Thus, nowadays, patients with ALN metastases at the time of the diagnosis are generally given neoadjuvant chemotherapy (NAC), aiming to achieve primary tumor (PT) shrinkage before surgery [5]. Pathological response on surgical specimens may be widely heterogeneous between PT and ALN, ranging from cases achieving complete response in both sites to others with significant residual neoplastic cells in breast or lymph nodes or both. Intratumoral heterogeneity is thought to be one of the main putative factors responsible for varying pictures of breast/nodal tumor response. This phenomenon results from the progressive accumulation of mutations in key oncogenes and tumor suppressor genes in normal mammary epithelium and cancer cells [6,7]. In this view, marker expression discordances and NAC response differences may be explained by the expansion and selection of specific neoplastic clones after administered therapies [8]. Moreover, in tumors with synchronous ALN metastases, the metastatic cells could represent the whole neoplasm’s biological recurrent and metastatic potential much better than the PT [9]. Therefore, in such cases, the expression of key biomarkers, including estrogen receptors (ERs), progesterone receptors (PRs), the Ki67 proliferation index, and human epidermal growth factor receptor-2 (HER2), should be assessed in both PT and metastatic ALNs. Discrepancies in biomarker profiles may reflect differences in clonal tumor populations and then influence the response to the available therapeutic regimens. Several works have focused on the different patterns of biomarker expression between primary BCs and synchronous ALN metastases in surgical specimens [10,11,12,13,14,15,16]. Although with variable rates, discordances in the biomarker profile between PTs and ALNs have emerged, such as hormone-sensitive tumors with negative metastases [10,11] or HER2-negative breast neoplasms with overexpression/FISH amplification in lymph nodes [13,14], among others.

Nevertheless, in such instances, previously administered therapies may impact the switch in biomarker profile, as clones in heterogeneous cancer cells resistant to drugs might grow and thrive. To date, the expression of BC biomarkers in both PT and ALN metastases at the time of diagnosis on bioptic material has not been explored, although it is theoretically more representative of the entire tumor biology. Thus, in the present study, we sought to investigate the profile of ER, PR, Ki67, and HER2 expression in primary breast tumors and synchronous ALN metastases in a series of preoperative bioptic samples, underlining relevant differences that are able to influence NAC, post-operative adjuvant strategies, and biological behavior.

## 2. Materials and Methods

### 2.1. Patients and Samples

A total of 301 consecutive core needle biopsy (CNB) samples of primary breast tumors matched with synchronous ALN metastases were retrieved from the archives of the Division of Pathology of the Humanitas Istituto Clinico Catanese (Catania, Italy). CNBs of both PTs and ALNs were performed within four years (2020–2023) using a 14-gauge Tru-cut needle tumor. As for current guidelines [17], based on tumor size, two cores per centimeter and up to six total samples were harvested for each specimen. Similarly, in the case of multiple PTs, each lesion underwent bioptic sampling. When more than one suspicious ALN metastasis was present instead, to avoid patient discomfort, the widest or most accessible lesion was biopsied to address therapeutic strategies. The study was conducted in accordance with the Declaration of Helsinki and approved by the Ethics Committee of Catania 2, A.R.N.A.S. Garibaldi-Nesima of Catania, Palermo Street 636, 5-95122, Catania (protocol code 3647/DSA, 30 December 2020). All patients gave their written informed consent to diagnostic procedures and treatment according to the institutional rules for everyday clinical practice and experimental evaluations on archival tissue. All the slides were reviewed by two experienced breast pathologists. As for morphological features, PTs and ALN metastases were both evaluated for their histological classification, nuclear tumor grading according to the WHO 2019 classification, and the presence/absence of necrosis.

### 2.2. Immunohistochemistry (IHC) and Fluorescence In Situ Hybridization (FISH)

Sections from tissue blocks of all the included PTs and ALN metastases were immunohistochemically stained with the following antibodies: ER (clone SP1, Ventana—Roche Diagnostics: Rotkreuz, Switzerland), PR (clone 1E2, Ventana), Ki67 (clone 30-9, Ventana), and HER2/neu (clone 4B5, Ventana). Tumors displaying apocrine morphological features were also tested with androgen receptors (clone SP107, Cell Marque: Rocklin, CA, USA) and GCDFP-15 (clone EP1582Y, Cell Marque). All samples were processed using a sensitive UltraViewTM Universal DAB Detection Kit (Ventana Medical Systems) detection system in an automated BenchMark ULTRA bond immunohistochemistry instrument (Ventana—Roche Diagnostics: Rotkreuz, Switzerland). Slices from both the PT and the ALN metastasis of the same cases were put on the same slide as long as the pathologist was separately provided with the original hematoxylin and eosin. Such a procedure was carried out to reduce the likelihood of bias due to technical issues occurring during each IHC run. The immunohistochemical expression of each marker for every tumor subtype was scored as follows, according to currently widely acknowledged guidelines: (i) ER/PR and Ki67 staining were recorded as the percentage of labeling cells, distinguishing between low (1–10%) and high cutoff rates (>10%) for the former [18] and between low (≤15%) and high levels (≥30%) for the latter [19,20,21], and (ii) HER2 immunolabeling was scored as 0, 1+, 2+, and 3+ as per the updated ASCO/CAP guidelines [22,23]. ER, PR, and HER2 immunostainings were carried out with properly validated positive controls. Cases that scored 2+ upon HER2 immunohistochemical analysis were reevaluated with FISH in order to assess *HER2* gene amplifications. FISH was performed on primary and metastatic samples using a dual-color Zyto Light SPEC ERBB2/CEN 17 probe (ZytoVision: Bremerhaven, Germany) and scored according to the ASCO/CAP guidelines [22,23] by experienced molecular biologists. HER-2 analysis was imaged and visualized using an AXIO Imager.M2 fluorescence microscope (Zeiss: Oberkochen, Germany), supported by the Neon-Metacyte Lite/Metafer (Metasystem: Altlussheim, Germany), to quantify neoplastic nuclei. Finally, biomarker discrepancies were also classified as clinically relevant and not relevant whether or not they were potentially able to modify therapeutic strategies according to the available BC treatment guidelines [5].

## 3. Results

Among the entire casuistry, patients’ age at diagnosis ranged from 32 to 84 years old (mean 59, median 58). Primary breast tumors measured from 0.4 to 7.5 cm (mean 2.6, median 2.3). As for histotypes, roughly 60% of the cases (183/301) were invasive breast carcinoma of no special type (NST), 15% (46/301) were invasive lobular carcinoma (ILC), and 13% (37/301) were mixed NST-ILC. The remaining 12% (35/301) were categorized as rare BC subtypes (mucinous, micropapillary, clear cell/glycogen-rich ductal carcinoma, metaplastic apocrine ILC, secretory, and cribriform, among others). With regard to nuclear grading, the neoplasms of the cohort were distributed as follows: 10% G1 (30/301), 67% G2 (202/301), and 23% G3 (69/301). As far as the discrepant cases were concerned, within the whole series, 31 of 301 tumors (10%) showed relevant differences in biomarker expression between the PT and metastatic ALNs. All but one of them were women (F:M ratio 1:29), and their age at diagnosis spanned from 35 to 81 years old (mean 60, median 59). At the time of diagnosis, primary breast tumors ranged in size from 0.5 cm to 6 cm (mean 2.4 cm, median 2.2 cm). One patient (case 11) was found with two different PTs within the right breast. As far as PT histology was concerned, on CNB less than half of the tumors of the divergent samples (16/31, 52%) were classified as invasive breast carcinoma NST and three cases (10%) as pure ILC. In comparison, six neoplasms (19%) showed heterogeneous NST-ILC features. The remaining PTs (6/31, 19%) belonged to rarer BC categories, namely, mucinous ductal carcinoma (2 cases), micropapillary ductal carcinoma (1 case), clear cell/glycogen-rich ductal carcinoma (1 case), metaplastic ductal carcinoma (1 case), and apocrine ILC (1 case). With regard to nuclear grading, 21 PTs (68%) were classified as G2, whereas the other 10 were classified as G3 (32%). Coagulative tumoral necrosis was spotted in two cases (6%). Compared to the corresponding PTs, in 27 of 31 discrepant samples (87%), CNBs of ALN metastases showed either different morphological features (17/31, 55%) (i.e., histological classification, architectural growth pattern, presence of necrosis), nuclear grading (4/31, 13%), or both (6/31, 19%) (Figure 1). The clinical–pathological features of the divergent cases of the present series are summarized in Table 1 and fully detailed in Appendix A.

As for immunohistochemistry, several different patterns of expression were encountered between PTs and ALNs, often affecting more than the investigated biomarkers, some examples of which are illustrated in Figure 2. In detail, the majority of the cases (16/31, 52%) displayed a significant difference (>30%) in hormone receptor expression in PTs and ALNs, equally distributed between samples switching from low (1*–*10%) to high (>10%) (8/16, 50%) and from high to low (8/16, 50%) percentages of positive cells. Furthermore, six neoplasms (19%) revealed relevant Ki67 proliferation index discrepancies, with all but one patient (case 10) changing from low (≤15%) levels in PTs to high (≥30%) rates in ALN metastases. Finally, regarding HER2 evaluation, differences in immunohistochemical or FISH scoring were detected in almost 3 to 4 of the cases (23/31, 74%), with increased and decreased values in 19 (61%) and 4 (13%) tumors, respectively. Among the former, in five instances (16%), a switch from HER2 negativity in PTs (1+ IHC or 2+ without FISH amplification) to overexpression (3+ IHC) or FISH amplification in ALNs was achieved; in another 14 samples (45%), the HER2 status changed from 0/1+ IHC score in PTs to 1+ or 2+ in ALNs but failed to demonstrate gene amplification upon FISH analysis. On the other hand, HER2 low-ALN immunohistochemical scoring (0 or 1+) from original overexpressing (3+)/FISH-amplified or IHC equivocal (2+)/FISH-negative PTs was observed in four cases. To note, multiple discrepancies involving two (7/31, 23%) or more (5/31, 16%) of the investigated biomarkers were noticed in a noteworthy percentage of divergent samples (12/31, 39%) (Figure 3A).

Concerning the clinical meaning, in a noteworthy percentage of cases (12/31, 39%) the different expression profiles between PTs and corresponding ALN metastases were theoretically able to influence NAC and postoperative adjuvant therapy strategies (Figure 3B). Namely, negative switching of hormone receptors in ALN would have allowed patient 28 to be treated as triple-negative. Similarly, the immunohistochemical/FISH finding of HER2 overexpression/amplification in metastatic ALN compared to HER2-negative PT would have let five patients receive anti-HER2 drugs in their NAC regimens. On the other hand, significant ER positivity in ALN in case 9, otherwise considered triple-negative BC in PT, would have supported the administration of hormone therapy in an adjuvant therapy setting. Likewise, two patients showed a reduced HER2 score (score 1+) in ALN rather than in PTs (score 3+ or FISH amplified). This would have questioned the following employment of anti-HER2 drugs conjugated with chemotherapeutics in the case of complete pathological response in PTs but not in ALNs on post-operative surgical specimens. Finally, in the forthcoming era of HER2-low tumors, stronger HER2 staining (score of 1+ or 2+/FISH negative) in ALN metastases than in PTs (score 0) in three cases would have influenced patients’ chance to be administered anti-HER2 conjugates. The discrepancies for the investigated biomarkers are summarized in Table 2.

## 4. Discussion

Breast cancer is a highly heterogeneous disease, not only regarding the existence of various histotypes in different patients but also concerning the emergence of multiple clones within the same tumor. For instance, in primary breast neoplasms, discrepancies in biomarker expression between preoperative biopsies and surgical specimens have been reported, accounting for approximately 10% of cases for both Ki67 [24] and HER2 [25]. Poor staining and especially intratumoral heterogeneity have been claimed as the most likely explanations for such findings. Similarly, it is acknowledged nowadays that only a few neoplastic cells from the primary breast tumor gain the ability to invade blood vessels and metastasize [14]. Therefore, tumor cells from metastatic foci often exhibit a different biomarker expression profile than PTs, reflecting the progressive acquisition of mutations in key oncogenes and tumor suppressor genes [26]. Thus, both the National Comprehensive Cancer Network (NCCN) [27] and ASCO [28] suggest repeating surrogate immunohistochemical and/or molecular tests in newly detected metastases to find relevant differences from PTs that are able to influence therapeutic strategies. Likewise, intratumoral heterogeneity explains why neoplastic cells in metastatic ALNs could better represent the biological potential of the whole disease than those in PTs [13]. In the present work, we compared immunohistochemical and FISH findings between PTs and matched synchronous ALN metastases on preoperative biopsies in a broad series of BCs. Overall, we found significant discrepancies in biomarker profiles in 10% of cases. In detail, our analysis revealed a discordance rate of 5.3%, 2.6%, and 7.6% as far as ER/PR, Ki67, and HER2 expression were concerned, respectively, corresponding to 52%, 26%, and 74% of divergent cases, respectively.

Several other studies have already addressed this issue, showing widely variable results spanning from nearly overlapping [29,30] to significantly discordant profiles, sometimes with differences in biomarker status in almost half of the cases [11,14,31]. However, all the aforementioned papers evaluated post-operative specimens, often including patients who had previously received NAC. It is well known that systemic therapies can potentially promote changes in biomarker expression [32]; hence, the lack of administration of prior therapies may account, at least in part, for the generally higher PT-ALN agreement rates in studies excluding patients receiving NAC [29]. To the best of our knowledge, differences in critical biomarker expression between PTs and ALN metastases in pre-operative CNBs have not been evaluated yet. Thus, our work is the first to accurately portray primary and metastatic BC tumor cells in a naïve status on initial bioptic material, guiding therapeutic management before any systemic or surgical procedure is performed. Namely, based on IHC and/or FISH findings of HER2 overexpression/amplification in synchronous ALN metastasis, five patients could have received subsequent anti-HER2 NAC drugs, which would have otherwise not been administered based on the negative expression in corresponding PTs. Similarly, in another instance (case 28) negative ER staining in metastatic ALN compared to positive PT could have allowed the patient to be considered affected by a triple-negative BC. Moreover, subjects receiving anti-HER2 drugs in NAC for HER2 overexpressed/amplified BCs are currently further administered anti-HER2 conjugates if they do not achieve complete pathological response, theoretically based on lack of biological response to “pure” trastuzumab therapy [33]. However, as some of our cases since the initial diagnosis switched from HER2 overexpression/amplification in PTs to negativity in metastatic ALNs, would it be advisable to treat them with anti-HER2 conjugates in the adjuvant setting as well as in the case of residual viable cells in ALNs but not in PTs? Additionally, in a noteworthy proportion of the samples (5 cases), we found a significant increase in the Ki67 index in ALNs (≥30%) compared to PTs (≤15%). As high Ki67 levels in ALN metastases but not in PTs have been associated with shorter survival [11,34,35], such a finding could further help to identify patients who may need more aggressive subsequent therapies and closer clinical follow-up [20].

Speaking of novel therapeutic opportunities, is it worth mentioning that almost a quarter (3/13, 23%) of HER2-discrepant cases from our casuistry showed an increase in ALNs from a 0/1+ to a 1+ or 2+ (FISH negative) IHC score, belonging to the so-called HER2-low BC category. It is acknowledged that patients diagnosed with these latter types of tumors are unlikely to take advantage of traditional “pure” anti-HER2 agents in the adjuvant setting [36]. However, these patients could benefit from anti-HER2 drugs conjugated with chemotherapeutics in the case of a near recurrence (within six months), as well as ab initio metastatic patients affected by similar tumors that did not respond to previous lines of systemic therapy [37]. When comparing histological specimens from PTs and distant metastases, a shift from HER2-negativity to HER2-low expression has been reported much more frequently than the opposite trend [38]. Hence, despite ALN localizations still not being considered distant metastases, our findings, along with others’ evidence on surgically resected specimens [39], suggest that these patients may benefit from anti-HER2 conjugates during their follow-up.

The present study carries some intrinsic limitations. Firstly, despite being supported by robust results, this was a single-center study. Hence, it is undoubtedly advisable to confirm our findings in a broader multicenter study merging the casuistry of other certified BC units. Secondly, our data were collected retrospectively: to strengthen our findings, a specifically drawn prospective study is indeed warranted in the near future. Namely, it would be particularly worthwhile to compare findings on biopsies with matched post-operative surgical specimens whenever residual viable neoplastic cells are available after NAC. In this view, it could be possible to properly investigate intratumoral heterogeneity between PTs and ALNs. Furthermore, it would also be possible to analyze the response of such different components to the eventual previous administration of NAC regimens, especially if influenced by the discrepant biomarker profile on CNBs. In this context, it should also be stressed that any proposed modification of the therapeutic management should previously undergo discussion in local multidisciplinary tumor boards. In such cases, shared collegial approval always ought to be reached in order to balance the benefits of adding further medications with their potential collateral side effects.

In summary, although indeed affecting a limited number of patients, our results advocate for biomarker profiles to be routinely performed on both PTs and metastatic ALNs in BC patients, as discrepancies in their status are likely to bear extremely relevant therapeutic and prognostic meaning [11,40]. Due to the high volume of breast CNBs in pathology laboratories, it might indeed be claimed that biomarker profiling on both PTs and ALNs in each BC case may not be economically affordable. These considerations notwithstanding, a few key strategies may certainly help to figure this issue out and lower internal costs: (i) Our data underline that roughly 90% of discordant tumors revealed significant morphological differences (histotype, nuclear grading, necrosis) between PTs and ALN metastases, suggesting that accurate pathological examination is proficient in identifying cases worth further ancillary testing; (ii) slices from both the PT and ALN could be put on the same IHC slides, as long as pathologists are separately provided with the original single hematoxylin and eosin glass slides. Nowadays, fine-needle aspiration cytology is still correctly considered by acknowledged guidelines to be a reliable method for investigating ALNs suspicious for BC metastases [1]. However, in cytological preparations, subtle morphologic differences between PT neoplastic cells and metastatic nodal cells may not be readily appreciated, but, as mentioned above, they could underlie significant discrepancies in the expression profile of biomarkers. Furthermore, cytological material might sometimes not be perfectly suitable for immunohistochemical analysis due to intrinsic technical issues (e.g., three-dimensional clustering of cells, crushing artifacts, obscuring material, etc.) [41]. Thus, in this specific contest, our data and others’ experiences [42] suggest that CNB of suspicious lymph nodes is more advisable in order to perform proper ancillary tests on more easily analyzable histological material.

## 5. Conclusions

Breast cancer is a widely heterogeneous disease, often made up of various neoplastic clones with different metastatic potential. In our series, we demonstrated that a relevant proportion (10%) of BC cases may show discrepant biomarker (ER, PR, Ki67, and HER2) expression between PTs and metastatic ALNs in preoperative CNBs. Even though they involve a restricted number of patients and further specifically designed, broader works are indeed warranted, such differences carry great potential to influence therapeutic strategies, pathological response, and, ultimately, patient outcomes. Therefore, our data claim to routinely assess key biomarker expression profiles in preoperative BC CNBs on both PTs and ALN metastases in cases displaying significant morphological differences.

## Figures and Tables

**Figure 1 diagnostics-14-00259-f001:**
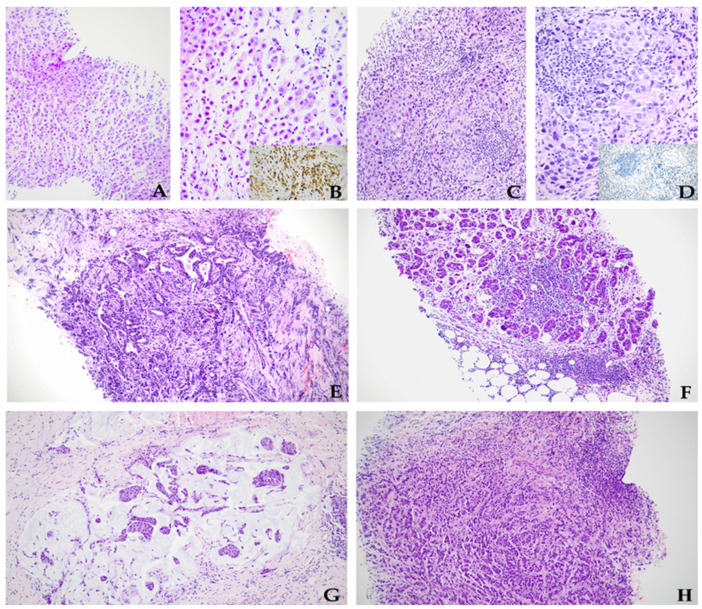
Hematoxylin and eosin pictures from the primary tumor (**A**,**B**) and axillary lymph node metastasis (**C**,**D**) of case 22, showing invasive lobular carcinoma with diffuse apocrine features in the former and heterogeneous conventional no-special-type/lobular carcinoma in the latter. The inset highlights GCPDF-15 staining, which was strongly positive in the primary tumor but negative in the nodal metastasis (see also Figure 2E,F for HER2). Case 24 revealed a no-special-type carcinoma in the primary tumor (**E**), contrasting with the micropapillary architecture of neoplastic cells within the axillary nodal metastasis (**F**). Microphotographs from case 27, displaying an invasive mucinous carcinoma in the primary breast tumor (**G**) opposite to a more conventional no-special-type architecture in the corresponding nodal metastasis (**H**). (**A**,**C**,**E**–**H**) are all at 100× magnification; (**B**,**D**) are at 200× magnification; inserts in (**B**,**D**) are at 400× magnification.

**Figure 2 diagnostics-14-00259-f002:**
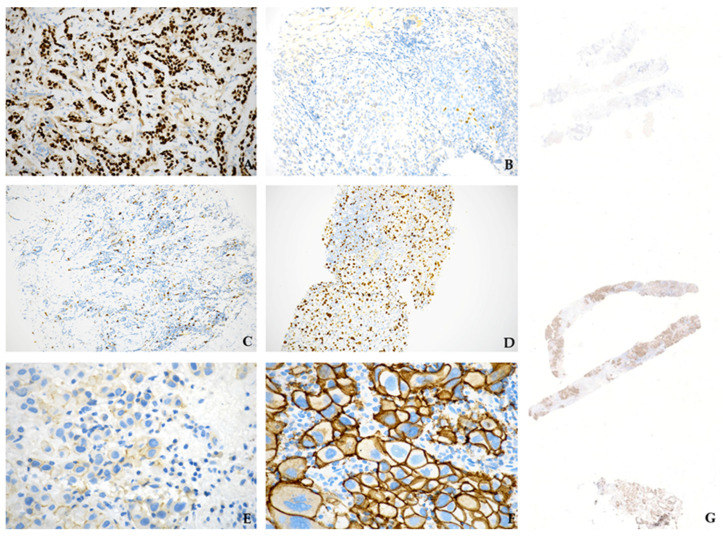
Discordant immunohistochemical findings between primary tumors and axillary lymph node metastases. Case 21 revealed diffuse estrogen receptor positivity in the breast primary tumor (**A**) compared to weak labeling (5%) in axillary lymph node metastasis (**B**); conversely, Ki67 proliferation index levels were lower in the primary tumor (**C**) than in the nodal metastasis (**D**). HER2 IHC from case 22 showed negative staining (1+) in the primary breast tumor (**E**), contrasting with overexpression (3+) by neoplastic cells in the metastatic lymph node (**F**); slices from the primary tumor, the nodal metastasis, and the positive control were placed onto the same glass slide for immunohistochemical analysis (**G**) (see also Figure 1A–D for morphology). (**A**–**D**) have 100× magnification; (**E**,**F**) have 200× magnification; (**G**) has 10× magnification.

**Figure 3 diagnostics-14-00259-f003:**
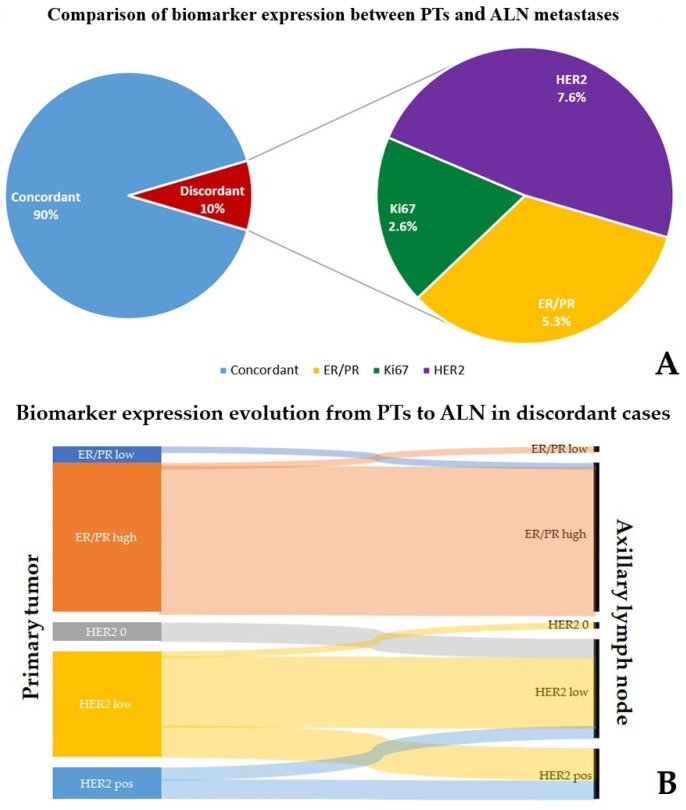
Comparison of biomarker expression between primary breast tumors and axillary lymph node metastases. Multiple discrepancies were observed in a relevant proportion of the discordant samples (12/31, 39%) (**A**). Sankey diagram showing the evolution of biomarker expression from primary tumors to metastatic axillary lymph nodes, many of which were able to influence therapeutic strategies (**B**). PT: primary tumor, ALN: axillary lymph node, ER: estrogen receptors, PR: progesterone receptor, HER2: human epidermal growth factor receptor 2.

**Table 1 diagnostics-14-00259-t001:** Clinical–pathological features of the divergent breast cancer patients in the present series showing biomarker profile discrepancies between primary tumors and axillary lymph node metastases.

Parameter	No. of Cases (%)	Parameter	No. of Cases (%)
**Sex**		**N. of lesions (PT)**	
F	29 (96%)	Single	29 (96%)
M	1 (4%)	Multiple	1 (4%)
**Age**		**Histology (PT)**	
35–81 y.o., median 59		NST	16 (52%)
≤59	13 (43%)	ILC	3 (10%)
>59	17 (57%)	Mixed NST/ILC	6 (19%)
		Other variant	6 (19%)
**Size**			
0.5–6 cm, median 2.2 cm		**Nuclear grading (PT)**	
≤2 cm	12 (43%)	G2	21 (68%)
>2 cm	16 (57%)	G3	10 (32%)
**Laterality**		**Necrosis (PT)**	
Right	23 (74%)	Absent	29 (94%)
Left	8 (26%)	Present	2 (6%)

Abbreviations: F: female, M: male, PT: primary tumor, NST: no special type, ILC: invasive lobular carcinoma.

**Table 2 diagnostics-14-00259-t002:** Summary of pathologically and clinically relevant discrepancies in biomarker expression between primary tumors and matched axillary lymph node metastases on core needle biopsies.

Pathologically Relevant								
**ER/PR**		**Ki67 (≤15% vs. ≥30%)**		**HER2**				
				**Increased**		**Reduced**		
Low to high	8	Low to high	5	0 → 1+		1	2+ (FISH neg.) → 1+	1
High to low	8	High to low	1	0 → 2+ (FISH neg.)		2	2+ (FISH neg.) → 0	1
				1+ → 2+ (FISH neg.)		11	3+ → 1 +	1
				1+ → 3+		3	2+(FISH +)/3+ → 1+	1
				2+ (FISH neg.) → 3+		1		
				2+ (FISH neg.) → 2+ (FISH +)		1		


Total	16	Total	6	Total		19	Total	4
**Clinically Relevant**								
**ER/PR**		**HER2**						
			**Increased**			**Reduced**		
Low to high	1		0 → 1+		1	3+ → 1+		1
High to low	1		0 → 2+ (FISH neg.)		2	2+ (FISH +)/3+ → 1+		1
			1+ → 3+		3			
			2+ (FISH neg.) → 3+		1			
			2+ (FISH neg.) → 2+ (FISH+)		1			

Total	2		Total		8	Total		2

Abbreviations: ER: estrogen receptors, PR: progesterone receptors, HER2: human epidermal growth factor receptor 2.

## Data Availability

The data presented in this study are available on request from the corresponding author. The data are not publicly available due to privacy restrictions.

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
