# Peer review of "Discordance of Biomarker Expression Profile between Primary Breast Cancer and Synchronous Axillary Lymph Node Metastasis in Preoperative Core Needle Biopsy"

_diagnostics, 2024, doi:10.3390/diagnostics14030259_

Round 1

Reviewer 1 Report

Comments and Suggestions for Authors

This is a well-written manuscript on differences in key predictive marker expressions in primary versus metastatic breast carcinoma. The illustrations are nice, the results are sound and well presented. I suggest the following points to be discussed.

- The weak sides of the study need to be discussed.

- Core needle biopsies are small samples of the tumor, discrepancies between the biomarker expressions on preoperative cores and operative specimens are well known, in some studies exceed the 10% regarded for relevant in the present work. 

- How many cores were taken and analyzed ?

- What if one has multiple primaries and multiple metastases ? 

There have been many publications during the last 50 years on differences in biomarker expression in primary tumor versus metastases (e.g. Pellas et al), thus being the first in this area cannot be true. Such statement is not needed in this correctly performed study. 

Author Response

We thank the Editor and the Reviewers for their comments, which substantially helped us improve our manuscript.

All comments have been taken into full consideration and the appropriate changes made.  All the changes are made and highlighted in red and colored in yellow, apart from English language editing.  Furthermore, the manuscript body has been expanded to reach the suggested total word count.

Please find below our responses to the reviewers’ comments.

Reviewer #1:

This is a well-written manuscript on differences in key predictive marker expressions in primary versus metastatic breast carcinoma.  The illustrations are nice, the results are sound and well presented.

We thank the Reviewer for appreciating our manuscript.

I suggest the following points to be discussed:

  • The weak sides of the study need to be discussed.

We thank the Reviewer for this comment.  A further paragraph about the study’s limitations has been added to the Discussion section of the revised manuscript.

  • Core needle biopsies are small samples of the tumor, discrepancies between the biomarker expressions on preoperative cores and operative specimens are well known, in some studies exceed the 10% regarded for relevant in the present work.

We agree with the Reviewer on this topic.  Several works have addressed the issue of biomarkers’ discrepancies in primary breast tumors between preoperative biopsies and surgical specimens, regardless of administered neoadjuvant chemotherapy: both for Ki67 (Ahn S et al. Arch Pathol Lab Med. 2018 Mar;142(3):364-368) and HER2 (Tamaki K et al. Cancer Sci. 2010 Sep;101(9):2074-9) an approximate divergence rate of 10% has been reported.  Poor staining and intratumoral heterogeneity are probably the main reasons for such a result.  Indeed, this latter is also likely to be involved in our findings of discrepant expression profiles between primary tumors and axillary lymph node metastases: accordingly, one of the main of our work was undoubtedly to investigate the different characteristics of the various neoplastic clones. Likewise, by biopsying both primary tumors and axillary lymph node metastases we sought to lessen phenotypic intratumoral variability due to cancer heterogeneity, catching at least those neoplastic populations able to influence patients’ management and therapeutic strategies

  • How many cores were taken and analyzed?

We thank the Reviewer for this suggestion.  As specified in the Material and Methods of the revised manuscript, according to current guidelines (Fusco N et al. Pathologica. 2022 Apr;114(2):104-110), at least two cores per cm of tumor for a maximum of 6 total cores were taken.

  • What if one has multiple primaries and multiple metastases?

We thank the Reviewer for this specification.  As documented in the revised manuscript's Material and Methods and Results, each of them was biopsied when multiple primary tumors were found.  Accordingly, one of the discordant patients (case 11) was found with two different primary tumors that, once biopsied, revealed similar morphological features but partially discordant biomarkers’ profiles.  Instead, in case of multiple suspicious nodal metastases, the largest or most accessible one was chosen to sample.  Such a decision was motivated by therapeutic reasons, as the pathological confirmation of just one positive node is enough to influence surgical and chemotherapy strategies and to avoid patients’ discomfort of undergoing repeated biopsies.

  • There have been many publications during the last 50 years on differences in biomarker expression in primary tumor versus metastases (e.g. Pellas et al), thus being the first in this area cannot be true. Such statement is not needed in this correctly performed study.

We thank the Reviewer for this advice.  We accomplish that various previous works have already addressed the topic of biomarkers’ profile discrepancy between primary tumors and axillary lymph node metastases.  Nevertheless, we wanted to underline that our study investigates it not on surgical specimens but on preoperative biopsies.  Accordingly, to avoid misunderstandings, the revised manuscript has been modified to highlight this point further and has also been provided with the suggested references.

Reviewer 2 Report

Comments and Suggestions for Authors

Marletta et al. explore theIHC and FISH profile of ER, PR, Ki67 and HER2 in a series of BC-matched primary tumors and axillary lymph node metastases pre-operative core-needle biopsies. They found 10% discrepancies in biomarkers’ expression between primary tumors and axillary lymph node metastases. The results hold considerable significance. Nevertheless, there are several unresolved matters that require further attention:

1.There are multiple errors in the abstract section of this article. “estrogen (ER)” should be “estrogen receptor (ER)”. And the English writing is not authentic enough. It is recommended to polish the entire article in English.

2.In the introduction section of this article, the marker expression discordances between primary tumors and axillary lymph node metastases needs to be re-described and the possible correlations need to be emphasized. In short, the focus of the introduction is not prominent enough.

3. In Table 1, I am puzzled why the number of cases for each feature is not the same? The author said they retrieved three hundred-one consecutive core needle biopsy (CNB) samples of primary breast tumors matched with synchronous ALN metastases in method. I think they should provide more information on the number of patients and subtypes in methods and tables.

4. In Figure3, discrepancies in biomarkers’ expression could be presented in the way of Sankey Diagram.

5. The authors said the discordance of biomarkers may influence therapies. Should they discuss in detail how the discordance of biomarkers between primary tumors and axillary lymph node metastases influence therapies? How should we select a treatment regimen in the presence of discordance?

Comments on the Quality of English Language

Moderate editing of English language required.

Author Response

Reviewer #2

Marletta et al. explore the IHC and FISH profile of ER, PR, Ki67 and HER2 in a series of BC-matched primary tumors and axillary lymph node metastases pre-operative core-needle biopsies.  They found 10% discrepancies in biomarkers’ expression between primary tumors and axillary lymph node metastases.  The results hold considerable significance.

We thank the Reviewer for appreciating our manuscript.

Nevertheless, there are several unresolved matters that require further attention:

  • There are multiple errors in the abstract section of this article. “estrogen (ER)” should be “estrogen receptor (ER)”.  And the English writing is not authentic enough.  It is recommended to polish the entire article in English.

We thank the Reviewer for the suggestion.  We checked the manuscript and had it edited by a native English speaker.

  • In the introduction section of this article, the marker expression discordances between primary tumors and axillary lymph node metastases needs to be re-described and the possible correlations need to be emphasized. In short, the focus of the introduction is not prominent enough.

We thank the Reviewer for this recommendation.  To underscore the focus of the present work further, the Introduction section has been enriched with an additional paragraph providing examples of biomarkers’ discrepancies documented in previous works.

  • In Table 1, I am puzzled why the number of cases for each feature is not the same? The author said they retrieved three hundred-one consecutive core needle biopsy (CNB) samples of primary breast tumors matched with synchronous ALN metastases in method.  I think they should provide more information on the number of patients and subtypes in methods and tables.

We thank the Reviewer for this comment.  Both in the text and the tables, we have reported the morphological and immunohistochemical/FISH findings of the 31 discrepant tumors of the entire 301 cases casuistry.  Captions of both revised Table 1 and Table S1 have been modified to further stress this point.  Regarding the different number of cases per feature of Table 1, as detailed in Table S1, they are mainly linked to the fact that one patient (case 11) was found with two distinct primary cancers so we overall had 31 tumors in 30 individuals.  Information about tumor size was missing for three patients, so we could report the corresponding data only in the remaining 28 cases.  Moreover, further details regarding the histopathological features of the whole series have been added to the Results section.

  • In Figure 3, discrepancies in biomarkers’ expression could be presented in the way of Sankey Diagram.

We thank the Reviewer for this suggestion.  Figure 3 has been modified according to the Reviewer’s request.

  • The authors said the discordance of biomarkers may influence therapies. Should they discuss in detail how the discordance of biomarkers between primary tumors and axillary lymph node metastases influence therapies?  How should we select a treatment regimen in the presence of discordance?

We thank the Reviewer for this recommendation.  As stated in the Discussion section of the revised manuscript, potential therapeutic implications are a key issue emerging from biomarkers’ discrepancy.  As detailed within the article, several therapeutic changes may be influenced by different expression of hormone receptors and HER2 status between primary tumors and nodal metastases (i.e. addition of anti-HER2 drugs or immune-checkpoint inhibitors in the neoadjuvant setting, administration of hormonal therapy in adjuvant regimens, etc …).  Of course, any modification of the therapeutic management should previously undergo collegial discussion in local multidisciplinary tumor boards in order to balance the benefits of adding further medications with their potential collateral side effects.

Round 2

Reviewer 1 Report

Comments and Suggestions for Authors

The manuscript is acceptable for publication after the revision.

Reviewer 2 Report

Comments and Suggestions for Authors

The author has fully addressed all my concerns. I recommend its publication.